# Polylactic Acid Composites Reinforced with Eggshell/CaCO$_3$ Filler Particles: A Review

Anahita Homavand [1], Duncan E. Cree [1,*] and Lee D. Wilson [2,*]

1 Department of Mechanical Engineering, University of Saskatchewan, Saskatoon, SK S7N 5A9, Canada; anahita.homavand@usask.ca
2 Department of Chemistry, University of Saskatchewan, Saskatoon, SK S7N 5C9, Canada
* Correspondence: duncan.cree@usask.ca (D.E.C.); lee.wilson@usask.ca (L.D.W.); Tel.: +1-306-966-3244 (D.E.C.); +1-306-966-2961 (L.D.W.)

**Abstract:** Statistics reveal that egg production has increased in recent decades. This growth suggests there is a global rise in available eggshell biomass due to the current underutilization of this bio-waste material. A number of different applications for waste eggshells (WEGs) are known, that include their use as an additive in human/animal food, soil amendment, cosmetics, catalyst, sorbent, and filler in polymer composites. In this article, worldwide egg production and leading countries are examined, in addition to a discussion of the various applications of eggshell biomass. Eggshells are a rich supplement of calcium carbonate; therefore, they can be added as a particulate filler to polymer composites. In turn, the addition of a lower-cost filler, such as eggshell or calcium carbonate, can reduce overall material fabrication costs. Polylactic acid (PLA) is currently a high-demand biopolymer, where the fabrication of PLA composites has gained increasing attention due to its eco-friendly properties. In this review, PLA composites that contain calcium carbonate or eggshells are emphasized, and the mechanical properties of the composites (e.g., tensile strength, flexural strength, tensile elastic modulus, flexural modulus, and elongation (%) at break) are investigated. The results from this review reveal that the addition of eggshell/calcium carbonate to PLA reduces the tensile and flexural strength of PLA composites, whereas an increase in the tensile and flexural modulus, and elongation (%) at break of composites are described herein.

**Keywords:** waste eggshell; calcium carbonate; PLA composite; mechanical properties





## 1. Introduction

Polylactic acid (PLA) is a common biopolymer which has gained interest as an attractive material of choice based on the current literature. The application of PLA as a biopolymer has progressively increased for different applications since it is considered eco-friendly, which is advantageous for industries that prefer to utilize materials that comply with environmental regulations [1–5]. PLA was developed for the first time in 1932 by Carothers, where this initial synthesis yielded a material with low molecular weight and limited mechanical properties. Further studies by Dupont in 1954 resulted in a higher molecular weight PLA, where it was later produced on a commercial scale in the late 1980s [6,7]. In 2002, PLA was synthesized for the first time at a larger industrial scale in the USA [8]. The addition of various low-cost fillers is a method to reduce overall fabrication costs, especially for PLA due to its higher value. The incorporation of mineral limestone (LS) or calcium carbonate (CaCO$_3$) with PLA has been shown to have a reinforcing effect on the composite matrix [5,9]. Chicken eggshells have been reported as a possible alternative to mineral calcium carbonate, since eggshells contain 96–97% CaCO$_3$ [10,11]. Statistics show that the annual global egg production has increased, attributed to an increasing global human population. Around 30% of eggs produced are sent to breaking plants where the eggs are cracked/broken for the separation of the hard shell and the liquid eggs. Therefore, breaking plants are the main sources of waste eggshells (WEGs), rather than homes and

restaurants, which may also present challenges for efficient collection [12,13]. The majority of these waste eggshells are disposed of in landfills at a cost to the breaking plant and can cause environmental problems [14,15]. However, waste eggshells can be consumed as an additive for human and animal consumption [14,16], soil amendment, cosmetics, catalysts, and sorbents [17–19]. The application of WEGs as a filler/reinforcement for the fabrication of PLA composites is a relatively novel method for the manufacturing of new products rather than discarding them as waste. In this review paper, an emphasis is placed on the use of mineral calcium carbonate and eggshells as fillers into PLA, and an overview of PLA composites that contain these two filler types is provided.

## 2. Biopolymers

Biopolymers are derived from natural substances, such as plants, animals, and bacteria [20,21]. For instance, biopolymers derived from plants are based on cellulose, lignin, starch, and alginate, while silk, wool, chitin, chitosan, and collagen are examples of natural biopolymers derived from biomass. Poly(hydroxyalkanoates) (PHA) and poly(hydroxy-butyrate) (PHB) are biopolymers fabricated from microorganisms during a microbial/bacterial fermentation or reaction [22–24]. As a result of global environmental concerns, the industrial demand for biopolymers fabricated from natural materials has increased. Due to their characteristics of biodegradability, non-toxicity, and renewability, the application of these biomaterials has expanded to various sectors such as automotive, biomedical, food/beverage, textiles, and agriculture [25–27]. Biodegradability is a term used to identify a polymer that can break down under relevant environmental conditions (temperature, pressure, etc.) and time [24]. Biopolymers are divided into two different groups: thermoplastic and thermosets. Thermoplastic biopolymers can be reshaped and remolded with heat and pressure [28,29], while thermosets cannot be reshaped after their curing cycle [28,30]. One of the important advantages of thermoplastic polymers is that they contain minimum chemical changes during and after processing since there is no curing cycle, in contrast to the production process of thermosets [28,31]. In this review paper, PLA as a thermoplastic biopolymer is considered.

Despite the advantages of biopolymers, there remains a challenge to reduce the current high cost of production and processing. Generally, biopolymers are more expensive to process and manufacture compared to conventional synthetic polymers, such as polyethylene (PE) and polypropylene (PP). Research has focused on different fabrication methods and strategies to reduce biopolymer production costs in an effort to make them economically viable replacements for conventional polymer materials. Recently, the number of studies to produce biopolymers from agricultural waste, food waste, and algae have increased, with a focus on reducing their fabrication costs [20]. PLA is among the most attractive biopolymers investigated [5,32] since this polymer is produced from renewable feed stocks from agricultural resources [23].

*Polylactic Acid (PLA)*

Lactic acid (LA) is the most common carboxylic acid, where it was discovered for the first time in 1780. There are two main processes for the fabrication of LA: chemical synthesis and microbial fermentation processes. Due to environmental concerns, the microbial fermentation process is the main method for producing LA [7,33]. In the beginning, LA was produced by the fermentation method was in 1881 by Avery Ltd. in the USA [7]. Corn and sugarcane are common raw materials used for producing LA [23,32], although wheat and potato peels are other agricultural resources that have been used to fabricate LA [34,35]. PLA is fabricated from the monomer of lactic acid during the fermentation process of sugar. Because of this, PLA is categorized as a biopolymer derived from plant-based substances [23,32].

Direct polycondensation and ring-opening polymerization are the two main processes that are applied to synthesize PLA. PLA with low to medium molecular weight

is produced with the polycondensation method, while PLA with high molecular weight (Mw $\geq$ 100,000 Da) is fabricated during the ring-opening polymerization method [36,37].

PLA has been used extensively in different industries since it is considered a suitable replacement for petroleum-based polymers, such as polyethylene terephthalate (PET), polystyrene (PS), and polypropylene (PP), as their mechanical properties are similar. Replacing conventional polymers with biodegradable PLA and increasing its application in various products (e.g., general applications, packaging, biomedical, etc.) can reduce the amount of non-degradable plastic waste in landfills and reduce the environmental impact of conventional synthetic polymers. One of the most important characteristics of PLA is its biodegradability into carbon dioxide, methane, water, and biomass fractions upon decomposition [38]. Due to the special characteristics of PLA, such as its biodegradability and good mechanical properties when compared to petroleum-based polymers, there is a high demand for this biopolymer, for diverse applications [36–38].

Various environmental conditions (e.g., pH, temperature, and moisture-levels) are key parameters which can affect the rate of PLA degradation [38–41], where greater temperature and pH enhance the rate of PLA degradation [38]. In a related study, Xu et al. [40] investigated the effect of pH on the PLA degradation rate, where the rate at pH 7.4 was four times higher, as compared to pH 3. Furthermore, upon increasing the temperature from 25 °C to 60 °C, PLA degradation occurred over 10 h versus 400 h at 25 °C. In addition, PLA degradation rate is faster in environments with elevated moisture, as compared to dry conditions, since PLA can absorb moisture/water which initiates the degradation of ester bonds of PLA via hydrolysis [36,42].

However, some properties of PLA have restricted its application, such as its brittleness and low heat distortion temperature [36]. The literature suggests that various techniques can be applied to overcome the drawbacks of PLA, such as blending, copolymerization, physical treatment, fabrication techniques of PLA composites, etc. In this study, fabrication of PLA composites is emphasized as a means to improve the properties of PLA [36,42].

## 3. Polylactic Acid (PLA) Composites

Polymer composites are materials which contain two or more components, where one component is the polymer and the second is either a filler or a reinforcement added as a component to improve the matrix properties. The addition of filler/reinforcement is a method to improve PLA mechanical properties. Furthermore, due to the lower glass transition temperature ($T_g$) of PLA, the application of PLA components/parts at high temperatures is limited; however, this challenge can be overcome by adding a filler/reinforcement [36]. For example, calcium carbonate or eggshell fillers added to PLA have been reported to improve tensile properties, $T_g$, and crystallinity of PLA [5,14,36]. On the other hand, filler/reinforcement particles tend to produce agglomerations, which reduce the PLA composites' mechanical properties [5,36]. Chemical treatments, such as stearic acid, oleic acid, and propionic acid of fillers (calcium carbonate, eggshell, etc.), are an appropriate method for decreasing the role of agglomeration and to overcome the reduced mechanical properties of PLA [14,36]. For PLA composites, there are different fillers/reinforcements that have been added, such as various types of natural fibers (flax [43,44], kenaf [45,46], and cellulose [47,48]) and filler particles, such as talc [49–51], mica [52,53], wood flour [51,54–56], rice husk [56], walnut shell [57], hydroxyapatite (HA) [57–60], calcium carbonate [5,61–64], seashell [45,65], and eggshell [5,9,66–68]. In this study, PLA composites reinforced with calcium carbonate and eggshells are emphasized as the source material.

## 4. Fillers

Fillers are added to the polymer composites to reduce the overall cost, although, in some cases, they can improve the properties of the polymer matrix. In the 19th century, fillers were added to polymers to produce composites at a lower cost [69]. There are two different types of fillers that have been added to PLA: natural [43–48,52–68] and synthetic [49–51] fillers, which are based on their origin. Fillers do not necessarily rein-

force polymer composites, although they can be considered reinforcements if mechanical properties are improved, as compared with the pure polymer [32,70,71].

*4.1. Natural*

The application and type of natural fillers added to PLA has expanded in recent years. These fillers enhance not only the properties of PLA but also meet environmental requirements. Composites with natural fillers are termed "green composites", and they reduce environmental concerns due to their biodegradability and green origins [70,72,73]. Natural fillers are divided into "mineral" and "organic" fillers. Mineral fillers, also known as inorganic fillers that are derived from mineral matter, such as rock and slag [74,75]. In comparison, organic fillers consist of organic materials and are found in irregular, acicular, fibrous, or flakey shapes, such as wood flour and seashell flour [76,77]. Mineral fillers, due to their wide availability/supply and relatively low cost, are added to polymers to improve their properties and reduce the fabrication cost by using a lesser amount of the more expensive polymer. Mineral fillers tend to have higher densities; therefore, the final product can be slightly heavier compared to products filled with organic fillers [74,77]. Calcium carbonate is a natural filler available in both forms of mineral and organic types [69,70].

*4.2. Synthetic*

Synthetic fillers are added to polymers for the production of composites with suitable properties for applications in the automobile, aircraft, and marine industries. In these fields, materials with improved mechanical properties are preferred, and synthetic fillers have some advantages, such as mechanical property improvements, over natural fillers. Various synthetic fillers, such as silicon carbide (SiC) [78–80], aluminum oxide ($Al_2O_3$) [80–82], zinc oxide (ZnO) [83,84], magnesium oxide (MgO) [85–87], graphite (although graphite is found in two types of natural and synthetic) [88,89], zeolite [90–92], etc., which have been added to polymers. These synthetic fillers are rigid in nature, and they improve the mechanical properties of the final polymer composites as well as reduce the cost of composites [89,93].

## 5. Calcium Carbonate ($CaCO_3$)

Calcium carbonate ($CaCO_3$) is one of the most important fillers added to PLA polymers. Addition of $CaCO_3$ to PLA have been reported to contribute enhanced heat resistance, stiffness, and hardness of the pure PLA. Furthermore, calcium carbonate is a common and cheap natural filler that has been extensively used in the industry for the design of polymer composites [4,63]. Another benefit of adding calcium carbonate as a filler to polymer composites, such as PLA composites, is to reduce the fabrication cost since a portion of the more expensive PLA is replaced with a cheaper filler [43,69]. For example, Prime Natural 4043D PLA (Jamplast Inc., Ellisville, MO, USA) costs approximately $5000 US per ton, and mineral limestone commercially sells for about $485 US per ton in North America [94,95]. Based on 2023 cost values, the reduction in PLA material cost using 10 wt.% calcium carbonate as a replacement for PLA would be 0.46 $US/kg. The 10-fold difference in the cost of PLA versus mineral limestone filler contributes to economic advantages and significant cost savings for the manufacture of PLA composites at larger scales. Putting this into perspective, a component with a weight of 1 kg of pure PLA would cost about $5 US. By adding 10 wt.% mineral limestone to this formulation, the cost would be reduced to approximately $4.54 US. Additionally, manufacturers probably can procure eggshells (as a calcium carbonate replacement) at a negligible cost since eggshells are a often treated as a waste material that can be readily obtained from egg-breaking plants. However, depending on the location of the breaking plant in comparison to the polymer composite manufacturing plant, a techno-economic analysis (TEA) should be conducted to determine its overall feasibility. Due to the high demand for calcium carbonate, the annual worldwide production of this material is reported to be near 10 million tons [93]. The main physical properties of commercially available $CaCO_3$ applied as a filler material is shown in Table 1 [69]. The particle size of $CaCO_3$, loading content, and its dispersion

within the polymer matrix are vital factors that have been reported to affect the composite's final mechanical properties [4,63].

**Table 1.** Main physical properties of commercially $CaCO_3$ filler used in polymers [69].

| Property | Normal Range |
|---|---|
| Specific gravity | 2.7 |
| Average particle size ($\mu$m) | 0.5–10.0 |
| Specific surface area ($m^2/g$) | >1–10 |
| Powder density ($kg/m^3$) | 800–1700 |

There are three different crystalline structures for calcium carbonate: calcite, aragonite, and vaterite. Calcium carbonate with a calcite structure is the most common form applied in polymer composites, since this crystalline structure can improve the material characteristics, which is essential in the application of polymer composite materials [69].

*5.1. Mineral Calcium Carbonate*

The earth's crust consists of calcium carbonate, and this material is abundantly found throughout the world. The mineral form of $CaCO_3$ occurs in different rock types, such as chalk, limestone, and marble, which are composed of calcite [12,14]. Approximately 10% of sediments in the oceans consist of $CaCO_3$. Furthermore, $CaCO_3$ is also a vital component of biological systems, such as shells of marine organisms, pearls, and eggshells [96]. Chalk is defined as a fine microcrystalline material and its application as a writing tool dates back nearly 10,000 years. Limestone has a more compact structure than chalk and is sourced from biogenic rocks. Marble is a coarse-crystalline calcium carbonate that is formed when chalk or limestone are put under specified conditions of high temperature and pressure. Limestone is used in the production of cement/concrete, toothpaste, paint, etc. A major application of limestone as an engineering material is as a filler for polymer composites [14,96,97].

*5.2. Biogenic Sources of Calcium Carbonate*

Another source of calcium carbonate is obtained from biomineralization systems. This type of $CaCO_3$ is known as bio-calcium carbonate, and one of the most central origins of biogenic $CaCO_3$ is chicken eggshells. Eggshells consist of a high amount of calcium carbonate or limestone. Previous studies showed that approximately 96–97% of eggshells consist of calcium carbonate, where the remaining content is organic matter [12,14].

**6. Chicken Eggshells**

Since chicken eggs are consumed worldwide, the egg industry is globally distributed [13]. The shell is the outermost hard layer of the chicken egg that can be found in a white or brown color [14]. Figure 1 shows eggshell biomass waste color as white (Figure 1a) and brown (Figure 1b). According to the reviewed literature, the amount of calcium carbonate in the white and brown eggshells are considered equivalent [14,67].

Eggshells are used in different industries, which are divided into two main groups. The first one is feed manufacturers who use eggshells to prepare feed for chickens, pigs, pets, etc., since they have a rich source of calcium carbonate. The second industry is manufacturers who apply eggshells in non-food applications, such as the production of fertilizers, cement production, etc. [12,16]. A chicken egg contains 60% albumen, 30% yolk, and 10% shell [14,98]. The eggshell also contains minor elements of other substances, such as magnesium oxide (MgO), sulfur trioxide ($SO_3$), phosphorous pentoxide ($P_2O_5$), aluminum oxide ($Al_2O_3$), potassium oxide ($K_2O$), silicon dioxide ($SiO_2$), dichlorine trioxide ($Cl_2O_3$), and strontium oxide (SrO). The amount of each of these substances is variable but typically less than 1%, and it depends on the type of feed consumed by the chickens [14,99,100].

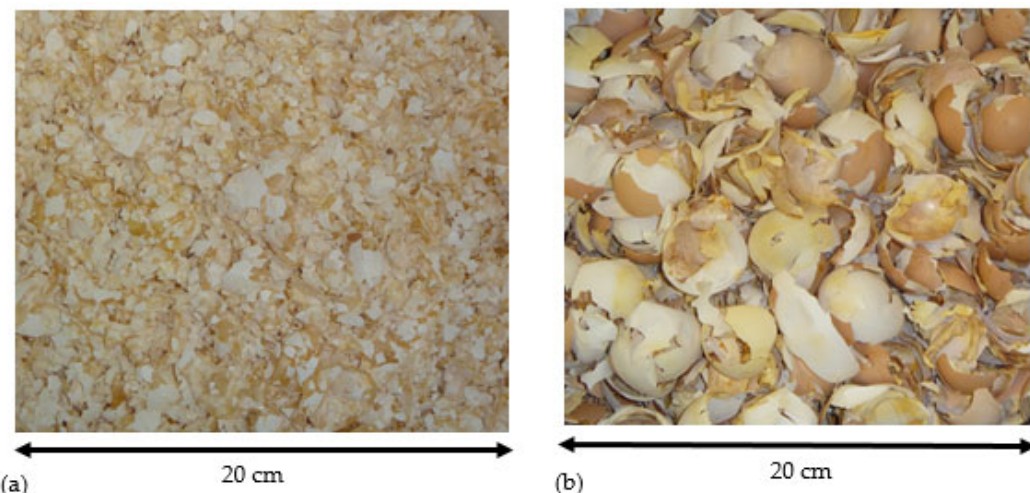

**Figure 1.** Eggshell biomass waste with variable shell color: (**a**) white and (**b**) brown.

*6.1. Global Egg Production*

The latest data published in 2023 for the global egg production show a continuous worldwide increase with an approximate growth of 3.5% from 1990 to 2021, as shown in Figure 2. According to the statistics, over a ten-year period (2011 to 2021), global egg production increased from 65.5 to 86.4 million tons [101]. This increase in egg production suggests greater availability of additional waste eggshell biomass. Finding solutions to repurpose this source of growing waste is a creative strategy required for future waste management [12,101].

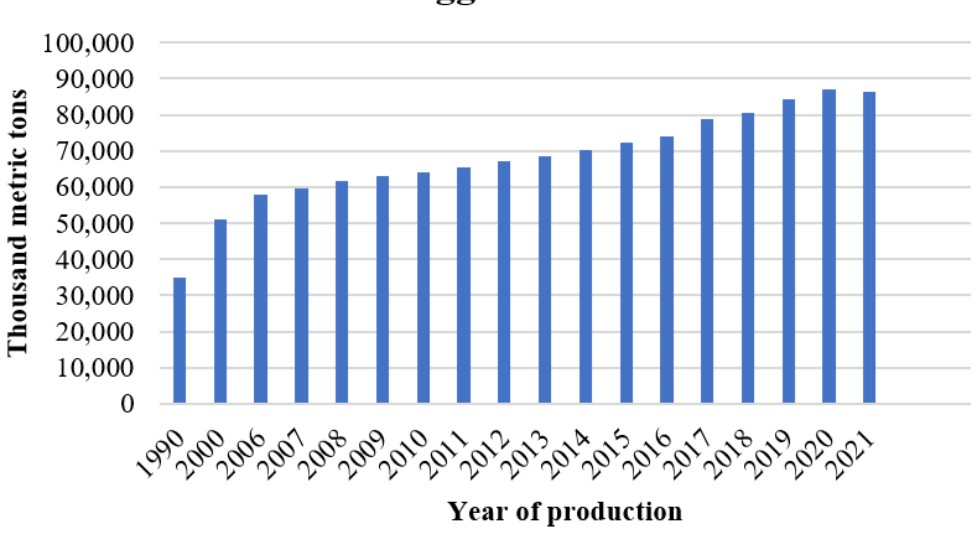

**Figure 2.** Global egg production from 1990 to 2021 in thousand metric tons [101].

According to Figure 2, there was a slight reduction in world egg production from 2020 to 2021. This has been justified due to the COVID-19 pandemic period, where a slowdown occurred in the production and distribution which affected the supply chains in all industries [101,102].

Furthermore, the maximum growth rate in egg production is reported to be in Asia, since China was the largest egg producer in 2021. Also, the second and third place egg producers are India and Indonesia due to the population density of these countries. Figure 3 shows the top ten leading egg-producing countries worldwide in 2021 [103].

## Leading egg producing countries in 2021

**Figure 3.** The top ten leading egg-producing countries worldwide in 2021 [103].

Table 2 shows differences in egg production for the top ten countries in 2019 and 2021. The leading countries were the same in 2019 and 2021. During the COVID-19 pandemic period, most countries experienced a reduction in their egg production, while other countries, such as India and Pakistan, raised their egg production [103,104].

**Table 2.** Comparison of the egg production of top ten countries in 2019 and 2021 [103,104].

| Countries | Number of Egg Production in Billions | |
|---|---|---|
| | **Year 2019** | **Year 2021** |
| Peoples Republic of China (PRC) | 636.82 | 586.31 |
| United States of America (USA) | 147.85 | 110.73 |
| India | 127.31 | 122.00 |
| Indonesia | 104.79 | 114.58 |
| Brazil | 69.62 | 58.20 |
| Mexico | 65.03 | 57.49 |
| Japan | 58.19 | 42.90 |
| Russia | 54.92 | 44.58 |
| Turkey | 27.41 | 19.30 |
| Pakistan | 19.05 | 21.28 |

Large quantities of waste eggshells are generated at egg-breaking plants, an industry where liquid white and yellow biomass from eggs are extracted. The waste eggshells are generally sent to landfills at a cost to the industry [12–14]. Therefore, finding new applications represents an innovative solution to add value to these agricultural wastes, which would ultimately divert them from disposal. Due to the increasing world egg production and the related high volume of waste eggshells, a number of studies have evaluated the application of these eggshells as an alternate source of calcium carbonate for various industries [14,16].

Eggshell Applications

There are several applications of eggshells to obtain value-added materials [16].

(1) Food additive for human nutrition

Eggshells are a rich source of calcium without any toxic substances, such as lead (Pb) and cadmium (Cd); therefore, WEGs can be an ideal option as a human dietary

supplement [16,105]. Furthermore, the eggshell membrane can be used as a rich nutritive food source to preserve healthy joints and connective tissues, since the eggshell membrane contains edible proteins [106,107].

(2)    Food additive for animal nutrition

The Association of American Feed Control Officials (AAFCO) has accepted eggshells as a feed additive for both companion and livestock animals. The most important point is the eggshells need to be dried at 80 °C before being added to animal foods. The drying process is applied for moisture and microbial removal [16,106].

(3)    Soil amendment

Due to the high amount of calcium carbonate in eggshells, this material can supply the soil with calcium mineral species, which is important for soil pH and fertility [16,18].

(4)    Purified calcium carbonate

Purified calcium carbonate can be used in the construction sector, i.e., as a building material, an ingredient in cement production, or in mortar/concrete formulations [16,67]. Also, it can be used in the paper industry to give brightness to paper. Furthermore, it can be used in the paints and dyes industries [108,109].

(5)    Cosmetics

Collagen is a key ingredient in many cosmetic products, specifically, in skin products to treat wrinkles and as an anti-aging effect. Eggshell membrane is a rich supplement of collagen, so this can be used in the cosmetics sector. In addition, some studies have shown that collagen extracted from eggshells is safer (e.g., reduced allergic reactions in cosmetics) than other collagen resources [16,110]. Moreover, eggshells are good facial cleansers and scrubs [106].

(6)    Catalyst and sorbent

Eggshells can be used as a catalyst in different reactions, such as biodiesel production [16,111,112] and dimethyl carbonate synthesis [113]. Also, eggshells can be used as a sorbent for organic and inorganic pollutant removal from water, wastewater, soils, or flue gas effluents [16,114].

(7)    Polymer composites

One of the applications of eggshells is in polymer composites, which has been applied as an alternate source of calcium carbonate filler, due to the calcium carbonate content (96–97%) of eggshells [12]. Polymer matrices have been reinforced with mineral calcium carbonate for different applications, such as for the automotive industry, biomedical applications, food and beverage packaging, and general-purpose plastic consumables (e.g., cup and fork utensils). Therefore, it is anticipated that eggshells can serve as a suitable replacement for mineral calcium carbonate for the preparation of polymer-mineral composites [5,36]. Figure 4 is a schematic diagram that illustrates the injection molding fabrication method of PLA composites that are reinforced with eggshells. This technique consists of two steps: eggshell preparation and composite fabrication. First, the as-received eggshells are washed several times to remove organic residues (albumen or egg white) from the eggshells and then dried at 105 °C for 24 h to remove residual water content. After this step, the eggshells are ground to achieve a specific particle size. Second, the eggshell particulates and PLA are mixed in a twin-screw extruder, and pelletized composite material is fed into an injection molding machine [115].

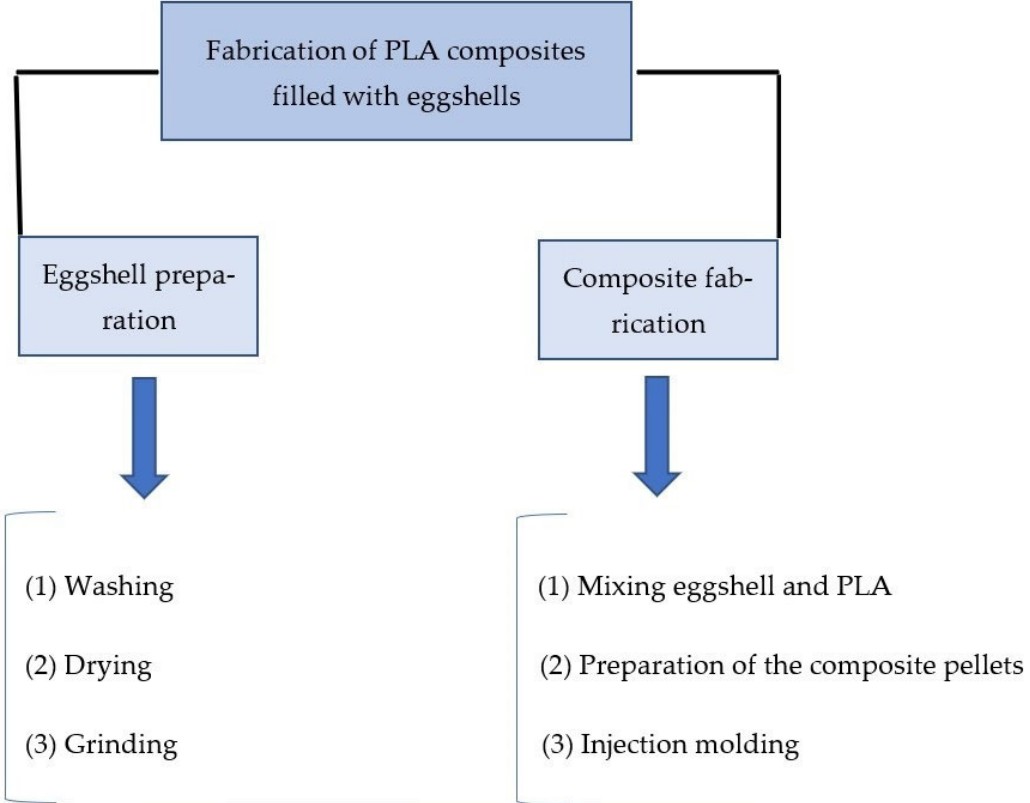

**Figure 4.** The fabrication of PLA composites reinforced with eggshell by the injection molding method [115].

## 7. Polylactic Acid Composites Reinforced with Eggshell/Calcium Carbonate Particles

Different characteristics of reinforcements/fillers affect PLA composite mechanical properties, such as the reinforcement/filler type, loading concentration, and particle size. In a related study, Betancourt and Cree [5] and Cree and Soleimani [115] investigated the effect of filler type (white eggshells and mineral limestone), loading amounts (5, 10, and 20 wt.%), and particle sizes (63 μm and 32 μm) to produce injection-molded PLA composites. Compared to the pure PLA, the addition of filler loadings of either limestone or eggshells caused tensile strength reductions for both particle sizes, where smaller particles (32 μm) showed a slightly higher tensile strength compared to larger (63 μm) filler particles. Furthermore, tensile strength values were greater for PLA composites filled with limestone compared to the composites filled with eggshells. According to the results, the highest tensile strength was observed for the PLA composite filled with 5 wt.% of limestone with a particle size of 32 μm. The tensile strength was reduced with an increase in filler loadings, since there was particle agglomeration with greater amounts of filler, which led to reduced tensile strength values. Also, by increasing the filler loading, tensile elastic modulus increased for both types of fillers (eggshells and mineral limestone) and both particle sizes. The maximum tensile elastic modulus was observed when 20 wt.% of the filler was added. Generally, the addition of a hard, rigid filler to a soft, less rigid matrix, results in greater composite stiffness overall. Also, flexural strength decreased with increased filler loadings, and the maximum flexural strength was observed for 10 wt.%. The reason for this phenomenon was reported to be similar to the tensile results. By increasing the filler loading, the possibility of agglomeration increases, and flexural strength decreases. The weak electrostatic and/or van der Waal interactions between agglomerated particles tends to be a less stable region of the composite material. According to the observations, PLA composites filled with limestone had a slightly higher flexural strength compared with the pristine eggshell fillers. Similar trends for the tensile elastic modulus results were observed for flexural modulus, where it improved with greater filler loading for both filler types

and particle sizes. In a related study, Cree et al. [116] investigated the effect of filler type (brown eggshells and mineral limestone), loading amounts (5, 10, and 20 wt.%), and particle sizes (63 μm and 32 μm) to produce injection-molded PLA composites. In this research, the effect of brown eggshell as a filler was investigated, compared to the previous study using PLA/white eggshell composites. The maximum tensile and flexural modulus was observed when 20 wt.% brown eggshell or limestone was added to PLA. Furthermore, by increasing the filler amount from 5 wt.% to 20 wt.%, the tensile and flexural strengths were reduced. Similar to the previous studies [5,116], mechanical properties, such as tensile and flexural strengths, as well as the modulus, were higher for PLA composites reinforced with 32 μm particles of brown eggshells or limestone. The studied mechanical properties were improved for PLA composites reinforced with limestone compared to the brown eggshell filler particles. Experimental mechanical tests, such as tensile and flexural strength, and the modulus, were reported in recent studies [5,115,116]. The results were compared using Analysis of Variance (ANOVA) F-test method as a statistical analysis, where the variable adjusted was the relative filler concentration. In a related study, Vigneshwaran et al. [117] developed a statistical linear regression model to predict the mechanical properties of PLA composites with an accuracy of 96%. This linear regression method helps to establish relationships between mechanical properties of different materials, and also aides in the prediction of mechanical properties for unknown materials.

In another study, Abdulkareem et al. [118] investigated the effect of calcined and uncalcined eggshells at variable loading levels (10, 20, and 30 wt.%) on the tensile properties of PLA composites, where the particle size was not reported. PLA composites were fabricated using a solution casting method. In this study, calcined eggshells were produced at 500 °C under ambient atmosphere. The results showed PLA composites filled with uncalcined eggshells had a maximum tensile strength when 30 wt.% filler content was added, while the highest tensile strength for calcined eggshell additions was for 10 wt.% content of filler. In another study, Hanumantharaju et al. [119] investigated the effects of different eggshell loadings (10, 15, 20 wt.%) on the mechanical properties of PLA composites (particle size not reported). In this research, PLA composites were fabricated by the 3D printing method of manufacturing. According to the results, the maximum tensile elastic modulus was observed when 20 wt.% eggshell was added to PLA, which was higher than the elastic modulus of pure PLA. By increasing the loading concentration of eggshells, the elongation (%) at break decreased compared to pure PLA. When 20 wt.% eggshells were added to PLA, the elongation (%) at break was less than that for pristine PLA. In a related study, Leclair et al. [120] investigated the effect of filler loading levels (5, 10, and 20 wt.%) for both white eggshell and limestone particles on the mechanical properties (flexural strength and modulus) of PLA composited fabricated by the 3D printing manufacturing method (filler particle size was reported as 32 μm). In this study, the determination of the optimal printing parameters for the flexural test was investigated. The best experimental results were at a print speed of 60 mm/s and a liquefier temperature of 200 °C. According to the results, the maximum flexural strength was obtained for PLA containing 5 wt.% white eggshell and 10 wt.% limestone. By increasing the filler loading amounts, flexural strength reduced and flexural modulus increased. In another study, Ashok et al. [68] studied the effect of eggshell powder loadings (1, 2, 3, 4, 5 wt.%) with a particle size of 25 μm on the mechanical properties of PLA composites fabricated by the film casting manufacturing method. The results showed that tensile strength and elastic modulus increased with eggshell content up to 4 wt.% and then decreased. With an increase in filler loading greater than 4 wt.%, agglomeration of eggshells occurred, which led to a decrease in the mechanical properties of the PLA composites. In addition, the maximum elongation (%) at break was observed when 1 wt.% of filler was added to PLA, which is lower than the amount of pure PLA. The elongation (%) at break increased slightly by increasing the filler loadings from 3 wt.% to 5 wt.%, although these amounts were lower than pure PLA.

*The Effect of Chemical Modification (Coating) of Eggshell/Calcium Carbonate Particles to Prevent Agglomeration in PLA Composites*

The agglomeration of fillers, such as calcium carbonate and eggshells, in polymer matrices is a critical challenge. The clustering of micro-sized filler particles causes a weak dispersion of fillers which reduces the mechanical properties of the overall polymer composite. To improve dispersion of calcium carbonate and eggshell fillers, a chemical coating (modification) applied to the surface of the particles has been suggested in the literature [62,121]. A coating can improve bonding at the interface between the PLA matrix and filler. Several studies have been conducted to evaluate the effect of chemical coating modifications of eggshell and/or mineral calcium carbonate particles on the mechanical properties of PLA composites. For example, Threepopnatkul et al. [121] studied the effect of modifying eggshell particles (particle size reported 150 µm, and 15 phr filler loading) with different types of fatty acids (e.g., propionic acid and oleic acid) on the mechanical properties of injection-molded PLA composites. The dispersion of eggshell particles modified with a fatty acid in a PLA matrix was observed to be more uniform than unmodified eggshells. Furthermore, tensile strength and tensile elastic modulus of the PLA/eggshell composites coated with propionic acid were higher than PLA/eggshell modified with oleic acid, although elongation and impact strength of PLA/eggshell modified with oleic acid are higher than pure PLA, non-modified PLA/eggshell, and PLA/eggshells that were modified with propionic acid. Table 3 shows a summary of the effect of different chemical treatment of eggshell on the PLA composite mechanical properties.

**Table 3.** Summary of the effect of different chemical treatments of eggshell on PLA composite mechanical properties [121].

| Type of Coating (Fatty Acid) | Eggshell Filler Loading (%) | Tensile Strength (MPa) | Tensile Elastic Modulus (MPa) | Elongation at Break (%) | Impact Strength (kJ/m$^2$) |
|---|---|---|---|---|---|
| Non-modified | 15 phr | 37.5 | 1310.0 | 3.8 | 2.1 |
| Oleic acid | 15 phr | 39.0 | 970.0 | 16.0 | 6.8 |
| Propionic acid | 15 phr | 41.0 | 1202.0 | 9.3 | 2.8 |
| - | 0 (pure PLA) | 54.2 | 1080.0 | 11.3 | 3.8 |

In a related work, Kumar et al. [63] studied the effect of mineral nano-calcium carbonate (particle size not reported) loadings of 1.5, 3, 5, and 7 wt.% on the mechanical properties of PLA nanocomposites produced by a solution blending manufacturing method that employed a solvent. The nanoparticles of calcium carbonate were treated with stearic acid. According to this study, there was an appropriate dispersion and a uniform distribution of nanoparticles within the polymer matrix. The authors suggested nano-sized stearic acid coated calcium carbonate particles provided a suitable candidate for the preparation of PLA nanocomposites.

Although the current review focused primarily on PLA composites, other biocomposites, such as bio-based polyethylene (PE) and bio-epoxy containing eggshells, have been evaluated. For example, Boronat et al. [100] developed a biocomposite based on bio-based PE and eggshells. In this study, the eggshell was chemically treated with titanate particles to increase filler (eggshell) adhesion to PE. In another study, Owuamanam et al. [9] investigated the fabrication and characterization of bio-epoxy eggshell composites. Even though the bio-epoxy had a 31% bio-content, being a thermoset, it was not biodegradable. The main benefit of using PLA instead of bio-based PE is the biodegradability property. PLA is a biodegradable material, while bio-based PE is not. Biodegradability is a vital characteristic when the material is used for specific applications, such as in the food and beverage industry, where biodegradability acts to reduce the packaging waste material after its one-time use [25–29].

### 8. Sustainability of PLA Composites

The automotive and food-beverage packaging industries tend to produce bio-based and sustainable materials built from PLA composites to overcome environmental challenges and concerns [122,123]. A recent application of PLA composites reinforced with $CaCO_3$ has been in an important biomedical application for the production of bone and tissue scaffolds [124]. In addition, three-dimensional (3D) printer filament is used in many industries [124] where a company (Filabot, Vermont, USA) recently introduced commercially available PLA/$CaCO_3$ filament for purchase by consumers [125]. PLA composites filled with eggshells are considered a sustainable approach, since such bio-based materials are made from renewable resources [12,20]. Another application of PLA/$CaCO_3$ composites were recommended for the production of gypsum boards as a building material to overcome noise pollution. The mechanical and sound properties of PLA/$CaCO_3$ composites were superior compared to the common gypsum boards and were found to be lighter in weight, which was suggested to be a possible alternative material to the conventional gypsum board [61].

Furthermore, the demand for producing bio-based materials and composites provide future guidance for studies that employ new environmental concepts for material design. For example, the use of Life Cycle Assessment (LCA) is an analytical method to enable the study of environmental effects of a product or process across the whole or partial life from the material extraction to the end of life [126,127]. In a related study, Ghomi et al. [127] investigated the LCA for PLA, where additional studies were recommended to further develop the LCA analysis for PLA composites. According to this study, the emission of greenhouse gases (GHGs) during PLA production in terms of carbon dioxide ($CO_2$) was lower, as compared with conventional petroleum-based polymers. The LCA of PLA showed that more than 50% of the released $CO_2$ was related to the conversion stage of bio-sources to lactic acid. By optimization of the PLA conversion process, there is a possibility to produce PLA with a lower carbon emission footprint.

### 9. Conclusions and Future Perspective

PLA as a biopolymer and as a composite material has recently attracted significant attention since PLA materials are considered eco-friendly. Many industries are focusing their products on materials which have minimum environmental concerns, which are driven by consumer demand. PLA production methods, such as direct polycondensation and ring-opening polymerization, can produce low to medium, and high molecular weight polymers, respectively. Fabrication of PLA composites by addition of various low-cost filler materials has been achieved, where the main goal enabled cost reduction in the overall composite. Furthermore, the level of global egg production continues to rise annually, which suggests that the quantity of waste eggshells is increasing. Finding innovative methods to use waste eggshells is a sustainable way to reduce this waste material and to alleviate related environmental concerns due to their disposal. One method is to add the eggshell as a filler to PLA polymers, where the filler and matrix are both sustainable materials. The studies in this review reveal that the replacement of mineral calcium carbonate fillers with an eggshell filler is a viable option. Different parameters, such as filler loadings, particle size, and chemical modification of fillers, were found to affect the mechanical properties of PLA composites. The smaller size and lower filler contents for both the eggshell and calcium carbonate resulted in improved mechanical properties. In general, the addition of eggshell and mineral calcium carbonate fillers decreased the tensile strength and flexural strength. By comparison, these fillers tended to increase the elastic modulus, flexural modulus, and elongation (%) at break. Reductions in tensile and flexural strengths at higher filler loadings were due to the role of agglomeration. In this condition, micro-sized filler particles tend to stick to each other through weak interfacial adhesion and produce agglomerations throughout the PLA matrix. In addition, by increasing the filler loading, tensile and flexural modulus increased, while the elongation (%) at break decreased. This was attributed to the rigid filler particles that were dispersed

in a flexible polymer matrix. Furthermore, the application of a chemical modification on the surface of eggshell and calcium carbonate particles serves to stabilize the interface between mineral and organic particles to prevent/reduce agglomeration of the fillers at higher contents. PLA/mineral calcium carbonate and/or eggshell composites have a wide range of applications that include biomedicine, the automotive industry, packaging, and also in general plastic consumer products such as utensils and children's toys.

Regarding future perspectives, more studies are required to investigate the effect of eggshells coated with different types and concentration levels of chemical additives, such as stearic acid, oleic acid, and propionic acid. Furthermore, the development of polymer nanocomposites is in its early stages; therefore, there is a current knowledge gap in the advancement of nano-eggshell PLA biocomposites. Due to the biodegradability potential of PLA in the human body, additional research is needed to advance the application of PLA composites that contain eggshell filler in critical applications, such as the fabrication of artificial organs. Another application of PLA composites filled with calcium carbonate relates to tissue engineering and to developing bone scaffolds, especially for bone generation. Additional studies consisting of PLA/eggshell composites should be developed for diverse applications in the biomedical field and on PLA composite LCA, since findings on these topics are in their early stages. We envisage that this review will catalyze further interest and inspire investigations related to polymer-eggshell composites.

**Author Contributions:** Conceptualization, A.H.; methodology, A.H.; formal analysis, A.H., D.E.C. and L.D.W.; investigation, A.H.; resources, D.E.C. and L.D.W.; data curation, A.H.; writing—original draft preparation, A.H.; writing—review and editing, A.H., D.E.C. and L.D.W.; visualization, A.H.; supervision, D.E.C. and L.D.W.; project administration, D.E.C. and L.D.W.; funding acquisition, D.E.C. and L.D.W. All authors have read and agreed to the published version of the manuscript.

**Funding:** The authors would like to acknowledge the financial support provided by Egg Farmers of Canada (EFC).

**Acknowledgments:** The authors would like to acknowledge the in-kind support of the industrial partners Agriculture and Agri-Food Canada, Egg Solutions EPIC Inc., Star Egg Company Limited, and Burnbrae Farms Ltd.

**Conflicts of Interest:** The authors declare no conflicts of interest.

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
