# Peer review of "Polylactic Acid Composites Reinforced with Eggshell/CaCO3 Filler Particles: A Review"

_waste, doi:10.3390/waste2020010_

Round 1
Reviewer 1 Report
Comments and Suggestions for Authors
Dear Authors,
This manuscript titled "Polylactic Acid Composites Reinforced with Eggshell/CaCO3 Filler Particles: A Review" provides valuable insights into the utilization of waste eggshells as fillers in polylactic acid (PLA) composites, offering a sustainable approach to waste management and composite material fabrication. The comprehensive review of worldwide egg production, applications of waste eggshells, and the mechanical properties of PLA composites filled with eggshell or calcium carbonate fillers is both timely and informative.
I believe that this review will significantly contribute to the field of waste utilization and sustainable materials development.
However, before publication, I recommend addressing a few queries:
It would be beneficial to include any limitations or challenges encountered during the fabrication and testing of PLA composites in your review.
Please ensure clarity and coherence in the discussion of the potential applications of waste eggshells, particularly in terms of their effectiveness as fillers in polymer composites.
How does the availability of eggshells relate to the global increase in egg production?
What is the potential benefit of adding eggshell or calcium carbonate fillers to polymer composites?
Why is polylactic acid (PLA) considered a high-demand biopolymer?
What specific mechanical properties of PLA composites filled with calcium carbonate or eggshells are investigated in the review?
Already published results may be compared using tables for clarity and comparision.
Some schematic diagrams may be used to explain the fabrication process involved in synthesis of polylactic acid composites reinforced with eggshell/CaCO3 filler particles.
How might the addition of eggshell or calcium carbonate fillers impact the fabrication costs of polymer composites?
Once these queries are addressed, I am confident that your manuscript will make a valuable addition to the journal.
Reviewer 2 Report
Comments and Suggestions for Authors
Review Comments for Manuscript
“Polylactic Acid Composites Reinforced with Eggshell/CaCO3 Filler Particles: A Review”
Dear Author,
Your manuscript presents a significant contribution to composite research. However, I would like to recommend a few suggestions for further enhancing the quality of your work. You are requested to make the following changes in the manuscript before publishing it.
- The abstract lacks specific mention of how eggshell/CaCO3 fillers impact the mechanical properties of PLA composites. Clearly define these effects to make the study's contributions effective to readers.
- The current literature review focuses primarily on PLA composites without offering a comparative analysis with other biopolymers. Expanding this section to include such comparisons will provide essential context and highlight the unique advantages of PLA composites.
- The manuscript does not clearly articulate the criteria for selecting the included studies. Detailing these criteria would enhance the research's transparency and allow for reproducibility of the methodology.
- There is a lack of statistical analysis in comparing the mechanical properties of PLA composites with different concentrations of eggshell/CaCO3 fillers. Employing statistical methods can validate the findings and strengthen the conclusions.
- The manuscript briefly mentions processing and performance challenges but lacks a detailed discussion. Addressing these challenges in depth will guide future research towards overcoming practical barriers.
- While the study hints at future research avenues, it does not specify them. Suggesting focused research directions, such as surface modification techniques for eggshell fillers, would provide a clear roadmap for enhancing PLA composites' performance.
- The environmental benefits of using eggshell/CaCO3 fillers in PLA composites are mentioned but not thoroughly assessed. Including a lifecycle assessment will quantitatively evaluate these benefits, reinforcing the study's sustainability claims.
- The manuscript lacks an in-depth discussion on PLA composites' scalability and economic viability with eggshell/CaCO3 fillers. Exploring these aspects will shed light on the commercial potential of these composites.
- The current references do not fully reflect the latest research developments in the field. Updating and ensuring the relevance of references will demonstrate the review's comprehensiveness and current relevance.
- Include case studies or examples of real-world applications of PLA composites with eggshell/CaCO3 fillers to illustrate the practical applications and benefits of the research findings.
Minor editing of English language required
Reviewer 3 Report
Comments and Suggestions for Authors
This review paper is good, and the review seems detailed and appropriate. Development of composite with biodegradable polymers is more exploratory as researchers from the community in this space are seeking alternative methods and approach for composite development. To make it better and more relevant to this research community it needs some additional details, especially in section 3 where more details on PLA with respect to its source synthesis will be helpful.
And in section 7 where details of more properties and structure evaluated and structure property relationship established will be helpful.
Comments on the Quality of English LanguageOverall English is good but minor structural changes are needed in sentence.
